# Kidney Cancer Biomarker Selection Using Regularized Survival Models

**DOI:** 10.3390/cells11152311

**Published:** 2022-07-27

**Authors:** Carolina Peixoto, Marta Martins, Luís Costa, Susana Vinga

**Affiliations:** 1INESC-ID, Instituto Superior Técnico, Universidade de Lisboa, Rua Alves Redol 9, 1000-029 Lisbon, Portugal; anacpeixoto@tecnico.ulisboa.pt; 2Instituto de Medicina Molecular—João Lobo Antunes, Faculdade de Medicina de Lisboa, Avenida Professor Egas Moniz, 1649-028 Lisbon, Portugal; marta.martins@medicina.ulisboa.pt (M.M.); lmcosta@medicina.ulisboa.pt (L.C.); 3Oncology Division, Hospital de Santa Maria, Centro Hospitalar Lisboa Norte, Avenida Professor Egas Moniz, 1649-028 Lisbon, Portugal

**Keywords:** kidney cancer, regularization, Cox regression, biomarker selection, gene ontology

## Abstract

Clear cell renal cell carcinoma (ccRCC) is the most common subtype of RCC showing a significant percentage of mortality. One of the priorities of kidney cancer research is to identify RCC-specific biomarkers for early detection and screening of the disease. With the development of high-throughput technology, it is now possible to measure the expression levels of thousands of genes in parallel and assess the molecular profile of individual tumors. Studying the relationship between gene expression and survival outcome has been widely used to find genes associated with cancer survival, providing new information for clinical decision-making. One of the challenges of using transcriptomics data is their high dimensionality which can lead to instability in the selection of gene signatures. Here we identify potential prognostic biomarkers correlated to the survival outcome of ccRCC patients using two network-based regularizers (EN and TCox) applied to Cox models. Some genes always selected by each method were found (*COPS7B, DONSON, GTF2E2, HAUS8, PRH2*, and *ZNF18*) with known roles in cancer formation and progression. Afterward, different lists of genes ranked based on distinct metrics (logFC of DEGs or β coefficients of regression) were analyzed using GSEA to try to find over- or under-represented mechanisms and pathways. Some ontologies were found in common between the gene sets tested, such as nuclear division, microtubule and tubulin binding, and plasma membrane and chromosome regions. Additionally, genes that were more involved in these ontologies and genes selected by the regularizers were used to create a new gene set where we applied the Cox regression model. With this smaller gene set, we were able to significantly split patients into high/low risk groups showing the importance of studying these genes as potential prognostic factors to help clinicians better identify and monitor patients with ccRCC.

## 1. Introduction

Renal cell carcinoma (RCC) is the most common type of kidney cancer [1]. In 2020, it was the 14th most common type of cancer (431,288 new cases), with a significantly higher incidence in developed countries, and the 15th most deadly one (179,368 deaths) worldwide [2]. These numbers of incidence and mortality are expected to increase to 666,000 and 301,000, respectively, by 2040 [3]. Among the different pathological subtypes of RCC, clear cell renal cell carcinoma (ccRCC) is the most common. ccRCC is known to have a significant percentage of mortality due to the high rate of metastasis and resistance to both radiotherapy and chemotherapy [4]. These, coupled with the fact that ccRCC early stage diagnosis is highly correlated to improved survival rate (83% at stage I vs. 6% at stage IV), make one of the priorities of kidney cancer research to identify RCC-specific biomarkers for early detection and screening of the disease [5]. This may help reduce both patient and healthcare systems’ burden due to kidney cancer.

Nowadays, there are no effective biomarkers for early diagnosis of RCC in clinical treatment, and the molecular mechanism of RCC metastasis remains unclear [6]. With the development of high-throughput techniques such as RNA-sequencing (RNA-seq), it is now possible to measure the expression levels of thousands of genes in parallel. This has been widely used for the discovery of new biomarkers in oncology [7], where the molecular profiling of individual tumors may help us find putative genes that will likely improve diagnosis and treatment outcome and guide patient selection for targeted therapies [8]. In particular, the relationship between gene expression and survival outcomes, such as the time to death, has been widely studied. Identifying genes associated with cancer survival may provide new information for clinical decision-making, diagnosis, prognosis, and treatment options of patients [7,9,10].

Survival analysis studies the time until an event of interest occurs, such as death. The Cox proportional hazards models have been used to discover attributes that are related to survival and predict the outcome [11]. These survival regression models describe the relationship between the survival times and a set of covariates [12] such as genes. Some studies show that molecular biomarkers may be more effective in predicting survival outcomes than clinical parameters such as tumor stage and grade [13]. Nonetheless, one of the main challenges of using gene expression data is their high dimensionality, since the number of covariates is much larger than that of observations. This can lead to instability in the selection of gene signatures, making the selection of novel biomarkers a difficult task [14]. Dealing with the high dimensionality of patients’ data represents a largely unsolved problem and several regularization methods have been proposed to tackle this challenge.

The Lasso penalty in the Cox model was proposed in 1997 [15], providing a solution with few variables selected. However, one of the problems of using this regularization is that when there are several correlated variables, Lasso randomly selects only one. More recently, to overcome this limitation, the elastic net that combines both Lasso and ridge penalties applied to Cox regression model was proposed [16,17]. The Lasso penalty chooses only a few nonzero coefficients, while a ridge regression scales all the coefficients towards zero [18]. Besides these methods, network-based regularizers have also been suggested. Since gene expression features may be connected through a graph structure where vertices are genes and edges a weighted relation between them, incorporating network information as a constraint in the loss function can be used to improve models’ performance and interpretability. Such network information can be either obtained by functional knowledge (e.g., protein–protein interaction network) available in public databases or a weight attributed to each gene based on data itself [9]. Other methods include using network information from gene coexpression [19] and also applying a network constraint based on gene correlation patterns between two groups of interest (e.g., normal and tumor) [20], moving towards more meaningful biological solutions.

Altogether, the application of Cox regularized models to survival data with RNA-seq covariates allows the identification of gene expression signatures and may enable the identification of targeted therapeutics and genes that can serve as predictive or prognostic biomarkers in kidney cancer [21].

Here we try to identify potential prognostic biomarkers correlated to the survival outcome of ccRCC patients. Several putative genes were found using network-based regularizers applied to Cox models and these were analyzed using gene ontology (GO) classification to try to find potential enriched mechanisms and pathways. GO (http://geneontology.org, accessed on 1 June 2022) is the most comprehensive and widely used knowledge base concerning the functions of genes, describing the biological role of genomic products (e.g., genes) by classifying them according to their molecular functions (MF), biological processes (BP) and cellular components (CC) [22]. These may be used to perform a gene set enrichment analysis (GSEA), which allows identifying over- and under-represented functional biological groups within a list of genes. We believe that these genes could serve as prognostic factors to help clinicians better identify and monitor patients with ccRCC.

## 2. Materials and Methods

### 2.1. Datasets

The data used in this study was retrieved from The Cancer Genome Atlas (TCGA) Research Network: https://www.cancer.gov/tcga (accessed on 1 May 2022) Both gene expression profile (RNA-seq) and clinical data from kidney renal clear cell carcinoma (KIRC) patients were used. Regarding RNA-seq, the initial size of the dataset was 20,501 genes for 606 samples, containing both tumor (n=529) and normal (n=77) tissue. After filtering genes with a null standard deviation, a total of 19,819 genes remained. Regarding clinical data, 537 samples had information on age, status (dead = 1 or alive = 0), days to death, days to the last follow-up, stage, T-stage, N-stage, M-stage, sex, and race (summarized in Table 1). The staging system used here was the American Joint Committee on Cancer (AJCC) TNM Classification of Malignant Tumors (TNM) system, which is based on the size of the tumor (T), the spread to nearby lymph nodes (N), and the spread to distant sites (M).

Data were downloaded from TCGA using the package RTCGAToolbox [23], which in turn processes RNA-seq with RSEM [24] that estimates gene and isoform expression levels. Data were then log transformed using the function $\log_{2}\left(x+1\right)$, which is a standard procedure when working with RNA-seq data to stabilize the variance across mean values [25].

### 2.2. Differential Gene Expression

A differential gene expression analysis was performed to assess which genes were differentially expressed between tumor vs. normal tissue and early stage (I, II, and III) vs. advanced stage (IV) of the disease, using the edgeR R package [26,27,28]. This package offers many variants of analysis. Here, we used the quasi-likelihood F-tests method, which is highly recommended for differential expression analysis of bulk RNA-seq data. The top ten most significant genes were examined and a multiplicity correction was performed by applying the Benjamini–Hochberg method on the *p*-values, to control the false discovery rate (FDR) [29].

### 2.3. Survival Analysis

A survival analysis studies the time until a certain event occurs (e.g., death). When analyzing survival data, two functions that are dependent on time are of particular interest: the survival function (S(t), the probability of surviving at least until time *t*), and the hazard function (h(t), the conditional probability rate of dying at time *t* having survived until that time). The Kaplan–Meier (KM, [30]) estimator is used to estimate the survival curve (S(t) against *t*) from a set of observed follow-up times and a variable indicating if the event took place or not [31]. The log-rank is a nonparametric statistical test used to compare the survival curves between two groups; it tests the null hypothesis that there is no difference between the population survival curves [32,33]. However, this test does not allow other variables to be taken into account. A Cox proportional hazards model [34] is a multiple regression model that allows the study of the relationship between several predictor variables and survival times.

#### Cox Regression

The Cox proportional hazards model is the most commonly used multivariate approach for analyzing survival time data in medical research. It describes the relationship between the event incidence (death) expressed by the hazard function and a set of covariates (genes) [12]. The Cox model can be expressed as: (1)$$h_{i}\left(t\right)=h_{0}\left(t\right)\exp\left(x_{i}^{T}\beta \right),$$
where $h_{i}\left(t\right)$ represents the hazard function of individual i=1,…,n, dependent on a set of *p* covariates $x_{i}={\left(x_{i1},x_{i2},\ldots ,x_{ip}\right)}^{T}$, and $\beta =(\beta_{1},\beta_{2},\ldots ,\beta_{p})$ are the regression coefficients. The $h_{0}\left(t\right)$ term represents the baseline hazard, the value of hazard if all $x_{i}$ are equal to zero. The inference of the β parameters is made by maximizing Cox’s partial log-likelihood function, given by: (2)$$l\left(\beta \right)=\sum_{i=1}^{n}\delta_{i}\left(x_{i}^{T}\beta -\log\sum_{j\in R_{i}}\exp\left(x_{j}^{T}\beta \right)\right),$$
where $R_{i}=R\left(t_{i}\right)=\left\{j:t_{j}\geq t_{i}\right\}$ denotes the set of all individuals that are at risk at $t_{i}$, i.e., with a follow-up time greater than or equal to $t_{i}$, and $\delta_{i}$ indicates if the event was observed ($\delta_{i}=1$) or not ($\delta_{i}=0$) for patient *i*.

To address the high-dimensionality problem (n≪p), sparse penalized Cox’s models have been considered, by adding a penalty term, F(β), to the partial log-likelihood, l(β). One example is an elastic net (EN), which uses the $\ell_{1}$-norm (Lasso) and the $\ell_{2}$-norm (ridge) to restrict the solution space by imposing sparsity and small coefficients to the parameters [16]. The EN penalty is defined as: (3)$$F\left(\beta \right)=\lambda \left\{{\alpha ∥\beta ∥}_{1}+\left(1-\alpha \right){∥\beta ∥}_{2}^{2}\right\},$$
where λ controls the penalizing weight and α the balance between the two norms (α=0 for ridge regression and α=1 for Lasso regression).

Network-based regularizers have also been proposed in the context of cancer genomics. The TCox method [20], a correlation-based regularizer, promotes the selection of features (genes) that have distinct correlation patterns in two different groups, such as tumor and normal tissue, highlighting potential differences in the corresponding subnetworks. Given two distinct datasets (tumor—T and normal—N), TCox builds the correlation matrices, $\sum_{T}=\left[\sigma_{1}^{T},\sigma_{2}^{T},\cdots ,\sigma_{p}^{T}\right]$, and $\sum_{N}=\left[\sigma_{1}^{N},\sigma_{2}^{N},\cdots ,\sigma_{p}^{N}\right]$, respectively. Each column $\sigma_{j}$ corresponds to the correlation of gene *j* with the remaining ones. The measure of gene *j* dissimilarity between T and N can be defined as: (4)$$d_{j}\left(T,N\right)=\arccos\frac{<\sigma_{j}^{T},\sigma_{j}^{N}>}{∥\sigma_{j}^{T}∥\;\cdot \;∥\sigma_{j}^{N}∥},j=1,\ldots ,p.$$

This dissimilarity term is then normalized by their maximum value:(5)$$w_{j}=\frac{d_{j}\left(T,N\right)}{\max_{k}d_{k}\left(T,N\right)},\;j,k=1,\ldots ,p.$$

In the context of the present study, we were interested in finding genes that exhibited different correlation patterns between tumor and normal tissues, i.e., genes that showed a larger dissimilarity between the two correlation matrices. Therefore, the penalty term used here is given by: (6)$$F\left(\beta \right)=\lambda \left\{{\alpha ∥q\circ \beta ∥}_{1}+\left(1-\alpha \right){∥q\circ \beta ∥}_{2}^{2}\right\}.$$
where vector $q=(w_{1}^{-1},\ldots ,w_{j}^{-1},\ldots ,w_{p}^{-1})$ represents the inverse of the normalized distances.

These models were built using the R package glmnet [35].

### 2.4. Model Evaluation

A survival analysis was performed using two different regularizations: EN and TCox. For both models, the α parameter was set between α=0.3 and α=0.05, which provided a feasible number of features to be further analyzed. All models were generated 100 times to ensure the robustness and stability of the results.

Samples were randomly divided into a training set (70%) for model construction and all data were used for the model evaluation. Both subsets had the same proportion of censored samples (≈67%). The training sets were used to find the best λ value (controls the penalizing weight) using a 10-fold cross-validation. This fixed λ was then used in the survival model applied to the whole dataset. To evaluate the accuracy of the models, the observations were split into two groups defined by the median of the fitted relative risks (high vs. low risk of dying). This allowed us to perform the log-rank test via the Kaplan–Meier estimator, and to assess if we could separate the survival curves of the two groups by evaluating the *p*-value.

After assessing the mean results obtained by each model for the 100 runs (Table 2), we observed that the best results were obtained when the parameter that controlled sparsity was set to α=0.1 for both the EN model and TCox. However, to select a similar and adequate number of variables to be further analyzed and interpreted in both regularization methods, gene sets obtained using TCox α=0.1 (≈51 genes) and EN α=0.2 (≈48 genes) were used to perform a further analysis. Notwithstanding, different α parameters may be tested to select different gene set sizes, using the code made available.

To perform the analysis described above, the glmnet [35] package was used in R statistical software. The q vector used in the TCox method was introduced as a penalty factor in the glmnet function.

### 2.5. Gene Set Enrichment Analysis

To highlight the enriched functions of biomarkers selected by the models tested, a gene set enrichment analysis (GSEA) was performed using the R package clusterProfiler [36]. The Gene Ontology (GO) [37] knowledge base is the world’s largest source of information on the functions of genes and defines classes used to describe gene function regarding three aspects: molecular activities of gene products (molecular function—MF), where gene products are active (cellular component—CC) and pathways made up of multiple gene products (biological process—BP). Given a ranked set of genes, GSEA determines whether the members of the gene set are randomly distributed or if they are primarily found at the top/bottom of the ranked list. There are three key elements of the GSEA method: the enrichment score (ES), which is the degree to which a gene set is over- or under-represented at the top or bottom of the ranked list, the significance level of ES, and the adjustment for multiple hypothesis testing.

GSEA was applied to four different gene sets to test how different metrics used to rank genes affected the selection of enriched ontologies. Firstly, we used as ranking measure the log fold change of DEGs found between tumor vs. normal tissue (GS1) and early vs. advanced stage of the disease (GS2). Afterward, genes were ranked based on β coefficients obtained from Cox’s regression model using an elastic net regularization (GS3) or TCox regularization (GS4). Then, the over-/under-represented GOs found were compared between groups, and genes that were more involved in enriched processes were further analyzed.

Lastly, KEGG (Kyoto Encyclopedia of Genes and Genomes) was used to perform a genes’ functional analysis [38,39,40]. KEGG is a collection of manually drawn pathway maps representing molecular interaction and reaction networks, covering a wide range of biochemical processes regarding metabolism, genetic information processing, environmental information processing, cellular processes, organism systems, human diseases, and drug development.

## 3. Results

Since molecular biomarkers may be a way of predicting the survival outcome of patients, studying the relationship between gene expression and survival phenotype may help us identify prognostic genes related to ccRCC survival, providing new information to help clinical decision making.

### 3.1. Exploratory Analysis

To identify putative biomarkers associated with survival outcomes in ccRCC patients, gene expression and clinical data (described in Table 1) from TCGA were used.

Firstly, a survival analysis using Cox proportional hazards model was performed to investigate the association between the time to death and the explanatory variables selected (stage, T-stage, N-stage, M-stage, sex, and race). In the multivariate Cox analysis, the only statistically significant covariates were the stages. In particular, the higher hazard ratio found was for stage IV, HR≈22 (*p*-value = 9.17 × 10^−5^), indicating a strong relationship between the patients’ stage and an increased risk of death.

Afterward, to find genes of interest associated with tumor formation and development in ccRCC patients, genes differentially expressed were found between tumor vs. normal tissues and also between the early (I, II and III) and advanced stages (IV) of the disease, using the edgeR package.

Comparing normal and tumor tissues, 13,906 differential expressed genes (DEGs) were found, 7076 up- and 6830 downregulated in tumor tissue. Table 3 shows the top 10 DEGs (ranked by FDR) found.

Furthermore, DEGs between the early and advanced stages of the disease were found. A total of 6170 genes were DEGs, 3126 up- and 3044 downregulated in the early stages of the disease. The top 10 genes with the lowest FDR values are listed in Table 4.

### 3.2. Survival Models

To find possible prognostic markers in kidney cancer, two survival models with different regularization metrics to handle high-dimensional data were used: EN and TCox.

#### 3.2.1. Elastic Net

Table 2 shows the results obtained for the Cox survival model with an EN penalty for the 100 runs tested with different α values, selecting a different number of variables. As we can see, the lowest *p*-value was obtained when α=0.1, selecting a mean of 90 genes (*p*-value = 0). However, to have a similar gene set size between the two methods used (EN and TCox), the genes that were selected as important in the Cox regression for at least 50% of the 100 runs using α=0.2 were listed (Table 5) and further analyzed, since those were the ones more correlated with the survival outcome of patients. A total of 50 genes were found. Some of these genes were upregulated and others downregulated (represented by ↑ and ↓, respectively, in Table 5) in tumor tissue or the advanced stage of disease. Here are presented only the top 20 genes selected. Three genes (*COPS7B, DONSON*, and *LOC100272146*) were always selected by EN.

#### 3.2.2. TCox

The second approach used to try to identify biomarkers associated with the survival outcome of patients was a penalization based on correlation—textttTCox. This regularizer promotes the selection of genes with distinct correlation patterns between the healthy and tumor tissues.

Table 2 presents the results obtained for this method for the 100 runs tested. The curves split with the lowest mean *p*-value was obtained for α=0.1, with a mean set comprising 47 genes. To assess which genes were more interesting to study, the ones selected at least 50% of the times were found (n=49). In Table 6 are listed the top 20 most selected genes and how often they are selected (represented by the percentage within the 100 runs tested). Five genes (*GTF2E2, HAUS8, PRH2, SEC61A2*, and *ZNF18*) were always selected by the model. Furthermore, comparing the two gene lists obtained for the methods used, four genes were found in common, *DONSON, SEC61A2, SNRPA1*, and *SORBS2*.

### 3.3. Gene Ontology

To describe a gene function along with the biological process, molecular function, and cellular component, a GSEA was performed. Gene sets were ranked based on their phenotypes using two different measures, the commonly used log fold change of DEGs and the β coefficient parameter of the regression. A total of four gene sets were tested: GS1—logFC of DEGs found in tumor vs. normal tissues, GS2—logFC of DEGs found between early vs. advanced stages, GS3—genes selected by EN method ranked by β parameter of regression, and GS4—genes selected by the TCox method ranked by the β parameter of the regression.

#### 3.3.1. DEGS Tumor vs. Normal

Firstly, the list of ranked genes based on the logFC between tumor and normal tissues was used (GS1). Enriched ontologies regarding biological processes (BP), molecular function (MF), and cellular component (CC) are presented in Figure 1, Figure 2 and Figure 3, respectively. For each analysis, enriched terms are listed on the left panel, and genes that belong to the top three enriched categories are shown on the right panel.

Looking closely at the BP characterization, we observed that the most significant ontologies were: urogenital system development (GO:0001655), kidney development (GO:0001822), and positive regulation of cytokine production (GO:0001819). Some of the DEGs associated with these ontologies showing a higher absolute logFC value were *PAEP, ORM1, SAA1*, and *LBP* (upregulated) and *AQP2, CALB1*, and *NPHS1/2* (downregulated).

Regarding the MF characterization, the top three ontologies found were immune receptor activity (GO:0140375), cation transmembrane transporter activity (GO:0008324), and inorganic molecular entity transmembrane transporter activity (GO:0015318), and the top DEGs found regarding the logFC were *AQP2, SLC12A1*, and *ATP12A*, all downregulated.

Concerning the CC classification, the top enriched ontologies were GO:0009897—external side of plasma membrane, GO:0045177—apical part of the cell, and GO:0016324—apical plasma membrane. *MUC17* was found to be one of the highest upregulated genes and *SLC14A1/2, ATP12A, UMOD*, and *AQP2* were downregulated.

Finally, regarding the KEGG enrichment pathways, the top three enriched terms were cytokine–cytokine receptor interaction (hsa04060), Human T-cell leukemia virus 1 infection (hsa05166), and chemokine signaling pathway (hsa04062).

#### 3.3.2. DEGs Early vs. Advanced Stage

Afterward, the same methods were applied to a gene set ranked by logFC between the early and advanced stage of disease (GS2) to see if different enriched functions and pathways with different gene sets were obtained.

Results for the BP classification are presented in Figure 4. The three most enriched ontologies found were mitotic cell cycle (GO:0000278), nuclear division (GO:0000280), and organic acid metabolic process (GO:0006082). Within these gene sets, *TAT, CRABP1, APOA1/5, APOC3, UGT2B4, SULT2A1*, and *ADH4* were upregulated in the early stages of the disease and gene *ANKFN1* was downregulated.

Regarding the MF classification (Figure 5), the main three ontologies were related to transporter activity (inorganic molecular entity transmembrane transporter activity—GO:0015318, transporter activity—GO:0005215, and transmembrane transporter activity—GO:0022857) and the genes found with the highest logFC between the early and advanced stage were *APOA1/2, SLC12A3*, and *AQP6*, all upregulated in the early stages of the disease.

Figure 6 presents the most significant ontologies regarding the CC characterization (plasma membrane region—GO:0098590, chromosomal region—GO:0098687, and secretory granule lumen—GO:0034774), and the genes involved in these terms. The genes with the highest logFC were *AHSG, TTR, HRG, ALB, AQP6*, and *LRRTM2*, all upregulated.

Finally, regarding the KEGG enrichment pathways, only one enriched term was found, the metabolic pathways—hsa01100.

#### 3.3.3. Genes Based on EN Regularization

To see how different metrics to rank genes affected the enriched gene ontologies and pathways found, we also used the β coefficient parameter of the regression to order genes.

Firstly, we used the β coefficients obtained from EN regularization (GS3). The three most enriched ontologies found using this ranked list of genes regarding BP characterization (Figure 7) were mitotic sister chromatid segregation (GO:0000070), nuclear division (GO:0000280), and sister chromatid segregation (GO:0000819). The genes with the highest correlation to the terms were *SMC4, KIF, NEK2, PLK1, TTK*, and *NUF2*, all with positive coefficients.

Regarding MF (Figure 8), the main three ontologies found were microtubule binding (GO:0008017), Ras guanyl-nucleotide exchange factor activity (GO:0005088), and tubulin binding (GO:0015631), and the highest correlated genes *KIF, PLK1, PRC1, MX2, GTSE1*, and *TPX2* were upregulated and *RGP1* was downregulated.

Figure 9 presents the most significant ontologies regarding cellular components, GO:0000775—chromosome, centromeric region, GO:0000776—kinetochore, and GO:0000793 —condensed chromosome. With *NEK2, NUF2, TTK, PLK1, CENPF*, and *KIF* being some of the genes with the greatest absolute value of the β coefficient, i.e., they have a higher impact on survival regression.

Lastly, regarding the KEGG enrichment pathways, the top three enriched terms found were Epstein–Barr virus infection (hsa05169), amyotrophic lateral sclerosis (hsa05014), and olfactory transduction (hsa04740).

#### 3.3.4. Genes Based on TCox Regularization

The last gene set analyzed (GS4) was ranked based on the β coefficients obtained for the survival regression using a TCox regularization.

Figure 10 shows the most enriched terms and genes found regarding the BP classification. The top three enriched terms found were mitotic sister chromatid segregation (GO:0000070), nuclear division (GO:0000280), and sister chromatid segregation (GO:0000819), the same as the ones found in GS3 (gene set ranked based on the β coefficients using EN regularization). Some of the genes with the highest β coefficient found were *NUF2, TRIP13, TTK, PRC1, KIF, NDC80, NEK2, PSRC1, AURKA*, and *FBX*.

Regarding the MF characterization (Figure 11), the top three enriched terms were microtubule binding (GO:0008017), tubulin binding (GO:0015631), and receptor regulator activity (GO:0030545), and the most important genes found were *NAV3, HAUS8, IFNB1, CCL8, GTSE1*, and *PSRC1*, all with positive β coefficients values. The *FNTA* gene was the only gene found in common between these three ontologies.

The top enriched terms regarding CC (Figure 12) were found to be GO:0000775—chromosome, centromeric region, GO:0000776—kinetochore, and GO:0000779—condensed chromosome, centromeric region. Furthermore, the genes with the greatest coefficient absolute value were identified as *NCAPG, NEK2, TTK, OIP5, NUF2, PLK1*, and *NDC80*.

Finally, the top three enriched terms regarding the KEGG enrichment pathways were found to be olfactory transduction (hsa04740), tight junction (hsa04530), and oocyte meiosis (hsa04114).

## 4. Discussion

To identify potential biomarkers correlated to the survival outcome of ccRCC patients, different approaches were tested, namely a differential gene expression analysis, network-based regularization applied to Cox models, and gene ontology classification. One of the limitations of using RCC TCGA data is that the information regarding cancer therapies is very sparse. Therefore, this confounder variable could not be considered, and we did not analyze how patients’ therapy may be associated with gene expression profiles and survival.

It has been shown that studying the relationship between gene expression and survival outcome is very important to identify genes associated with cancer survival, providing new information for the prognosis and treatment of cancer diseases. Here, we tested two regularization functions applied to the Cox regression: EN and TCox.

Regarding EN, a regularization method that combines penalties from both Lasso and ridge regression, we could see that some genes were always selected by the model in all of the runs tested (Table 5). One of those genes selected was *COPS7B* (COP9 Signalosome Subunit 7B). The upregulation of this gene has been associated with an advanced stage of the disease and metastasis in RCC, indicating that it may serve as a prognostic marker and therapeutic target [41]. Another gene always selected was *DONSON* (DNA replication fork stabilization factor). Previous studies have shown that the upregulation of this gene is associated with an advanced TNM stage and unfavorable prognosis in gastric cancer tissue [42]. Of note, this biomarker was previously associated with unfavorable overall survival in KIRC, and its prognostic potential was validated using quantitative real-time PCR and IHC, showing that this gene may be a robust biomarker for risk stratification with upregulation being associated with worst survival [43]. Here, both genes were found upregulated in the advanced stages and tumor tissue of ccRCC patients.

Afterwards, a penalization based on correlation—TCox was used. Within the genes found using this method (Table 6), some were also selected by the EN regularization (*DONSON, SEC61A2, SNRPA1*, and *SORBS2*). All of these genes were already described to have a role in tumor formation or metastization. For example, several studies identified new roles of *SNRPA1* in the progression of CRC, classifying it as a potential therapeutic target in the treatment of CRC [44] and showing that the upregulation of this gene was associated with a worst prognostic [45]. *SEC61A2* may be important for metastasis to the brain in humans when upregulated [46] and finally *SORBS2* has been reported to be a tumor suppressor gene in ccRCC [47,48]. It is worth mentioning that in this paper, although upregulated in tumor tissue, this tumor suppressor gene was found downregulated in metastatic patients (stage IV).

Regarding the TCox regularization, five genes were always selected by the model: *GTF2E2, HAUS8, PRH2*, and *ZNF18*. Interestingly, some genes were already associated with cancer disease. *GTF2E2* (general transcription factor IIE subunit 2) was found upregulated in lung adenocarcinoma tissue and was negatively associated with patients’ overall survival [49] and the genetic mutations of *PRH2* (proline-rich protein HaeIII subfamily 2) were also identified in lung cancer tissue [50]. Lastly, *ZNF18* is a zinc finger protein involved in transcriptional regulation [51] and *HAUS8* (augmin-like complex subunit 8) a microtubule-binding complex involved in the mitotic spindle assembly and maintenance of centrosome integrity [51], both processes related to enriched GO terms found here when we performed the GSEA (mitotic cell cycle, microtubule binding and chromosomal region).

As explained above, to better understand the enriched functions of genes selected, a GSEA was used to identify over- and under-represented functional biological groups (regarding BP, MF, and CC terms) within distinct list of genes. Four lists of genes were used: (1) DEGs tumor vs. normal ranked by logFC (GS1); (2) DEGs early vs. advanced stage ranked by logFC (GS2); (3) EN ranked by β coefficient (GS3); and (4) TCox ranked by β coefficient (GS4).

In Table 7 are listed some of the genes present on the top three enriched gene ontology terms found for each group tested (BP, MF, and CC).

When comparing gene ontology concerning the biological process (BP) terms between all gene sets tested, genes present in pathways involving nuclear division (GO:0000280) and sister chromatid segregation (GO:0000819) were found in common. Regarding the top genes present in these terms, we found genes involved in carcinogenesis with potential prognostic and therapeutic roles in cancer (*FBXO5* [52] and *SMC4* [53]) and also *PRC1*, whose deregulation is related to chromosomal instability and tumor heterogeneity [54].

The second gene ontology classification studied was the molecular function (MF), which describe activities that occur at the molecular level. Several enriched GO terms were found in common between the gene sets studied: various transmembrane transporter activities (GO:0008324, GO:0015318, and GO:0022857) and also microtubule and tubulin binding (GO:0008017 and GO:0015631). In Table 7 are listed the genes that contribute the most to these GOs. *SLC34A1* and *SLC12A3* that were found to be more involved in the transmembrane transporter activities terms have been studied as potential targets for the clinical diagnosis, prognosis, and treatment of ccRCC patients [55]. Regarding microtubule and tubulin binding, the *KIF* gene family was highly present in these terms. This superfamily contains microtubule-dependent molecular motor proteins, which, upon alteration of their expression, lead to cancer development and progression [56]. Likewise, *TPX2* has been studied as a factor critical for mitosis and spindle assembly and as a marker for diagnosis and prognosis when overexpressed in cancer [57]; *GTSE1* can also act as an oncogene and a high expression was positively correlated with histological grade and poor survival [58]. In this study, both genes were found upregulated in tumor tissue of ccRCC patients.

The last GO considered describes the locations relative to cellular structures in which a gene product performs a certain function—cellular component (CC). The most enriched terms found between the gene sets were related to plasma membrane regions (GO:0098590, GO:0045177) and chromosomal region (GO:0098687, GO:0000776, GO:0000779). Regarding plasma membrane regions, genes belonging to the urea transporter family (*SLC14*) were highly present and alterations in those genes may increase the risk of bladder cancer [59]. Further, the aquaporins (AQPs) family responsible for the transport of small solutes were also highly present, with known functions in tumor biology such as cell proliferation and migration [60]. Finally, the most significant genes belonging to chromosomal regions were found (*TTK, NEK2*, and *NUF2*). The *TTK* gene is associated with cell proliferation and proteins encoded by this gene are essential for chromosome alignment and duplication. Furthermore, it was found strongly overexpressed in human pancreatic ductal adenocarcinoma [61], which is consistent with the results obtained in this study, where we found *TTK* gene upregulated in the tumor tissue of ccRCC patients. *NEK2* and *NUF2* have also well established roles in cell cycle regulation and cell proliferation and their overexpression was associated with a variety of cancer types including renal cell carcinoma [62,63].

Altogether, we were able to find a set of putative genes that are correlated with survival outcomes in Renal Cell Cancer (*COPS7B, DONSON, SEC61A2, SNRPA1, SORBS2, GTF2E2, HAUS8, PRH2*, and *ZNF18*) and also a set of genes with some enriched ontology associated (*FBXO5, SMC4, PRC1, SLC34A1, SLC12A3, KIF, TPX2, GTSE1, SLC14, AQP, TTK, NEK2*, and *NUF2*).

To see if this smaller gene set may split ccRCC patients into two groups regarding their risk of dying, we created a new model where we applied a Cox regression using the ridge penalization (no feature selection was performed) to three different datasets, namely the full dataset (n=527), the early stage patients (n=441), and the metastatic/stage IV patients (n=84).

We were still able to split the curves with a mean significance *p*-value in the three groups of patients tested (Figure 13) and Table 8 shows the mean β coefficients obtained for each gene after we tested the model 100 times. Genes showing positive β coefficients (e.g., *COPS7B*), have HR > 1, so patients with an increased expression of these genes are expected to have a higher risk of dying. Likewise, genes showing negative β coefficients (e.g., *SORBS2*), have HR < 1, so patients with a downregulation of these genes are expected to have a higher risk of dying. Interestingly, as described before, *SORBS2* has been reported to be a tumor suppressor gene in ccRCC and here, we found that it was downregulated in metastatic tissue.

We observed that even though this gene set was able to split the two survival curves (high/low risk of dying) significantly, the hazard ratios obtained for each gene were close to one (HR=1, no effect). Therefore, the Cox regression with ridge penalization was applied only to the genes selected earlier by both EN and TCox regularizers (*DONSON, SEC61A2, SNRPA1*, and *SORBS2*). This was performed 100 times and Table 9 shows the mean β coefficients and HR obtained for each gene.

These results show that the biomarkers previously selected using the full dataset are also significant for risk stratification in early stage patients and also metastatic patients. This gene set comprises potential prognostic biomarkers with putative roles in cancer therapy that may help clinicians in the clinical decision-making of ccRCC patients.

## 5. Conclusions

One of the priorities of kidney cancer research is to identify RCC-specific biomarkers for the early detection and screening of the disease to reduce patient and healthcare systems’ burden due to kidney cancer. Nowadays, there are no effective biomarkers for early diagnosis of RCC in clinical treatment, and the molecular mechanism of RCC metastasis remains unclear.

Here, we identified a gene set of potential prognostic biomarkers correlated to survival outcome (using two different regularizers, EN and TCox) and enriched gene ontologies of ccRCC patients (*COPS7B, DONSON, SEC61A2, SNRPA1, SORBS2, GTF2E2, HAUS8, PRH2, ZNF18, FBXO5, SMC4, PRC1, SLC34A1, SLC12A3, KIF, TPX2, GTSE1, SLC14, AQP, TTK, NEK2*, and *NUF2*). Most of these genes were already described in the literature to be related to cancer formation and progression and a small set to kidney cancer, specifically.

Altogether, these genes should be further investigated as potential prognostic factors to help clinicians better identify and monitor patients with ccRCC.

## Figures and Tables

**Figure 1 cells-11-02311-f001:**
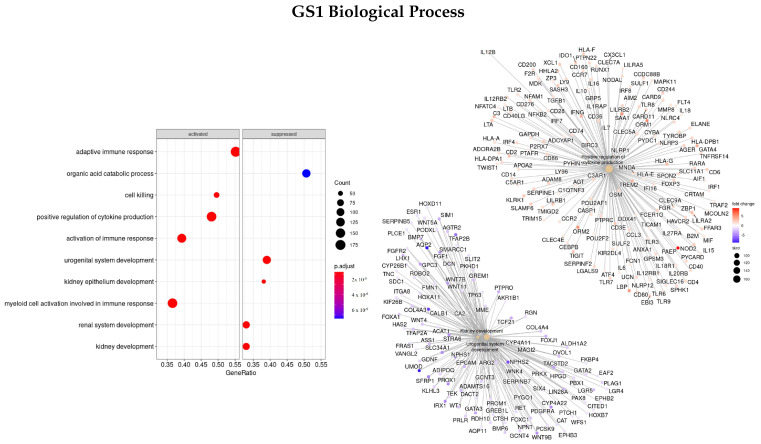
Gene ontology enrichment analysis regarding biological processes terms for a list of DEGs ranked by the log fold change between tumor and normal tissues. The left panel shows a dot chart with the most significant BP terms. The right panel shows a gene-concept network plot of the three most enriched terms that depicts the linkages of genes and biological concepts as a network.

**Figure 2 cells-11-02311-f002:**
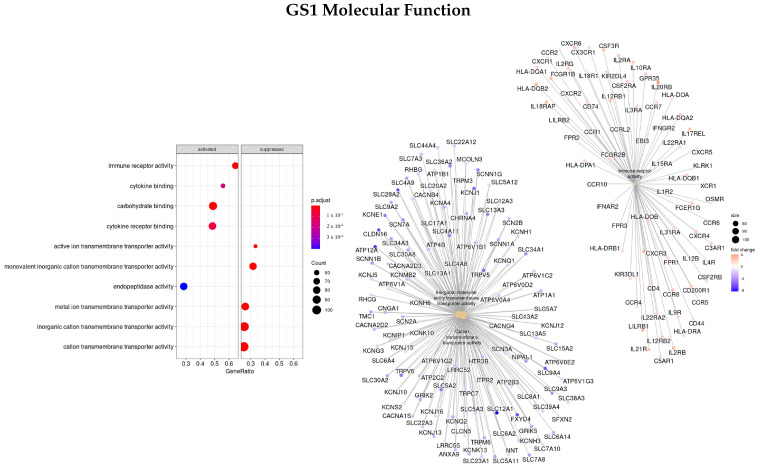
Gene ontology enrichment analysis regarding molecular function terms for a list of DEGs ranked by the log fold change between tumor and normal tissues. The left panel shows a dot chart with the most significant MF terms. The right panel shows a gene-concept network plot of the three most enriched terms that depicts the linkages of genes and biological concepts as a network.

**Figure 3 cells-11-02311-f003:**
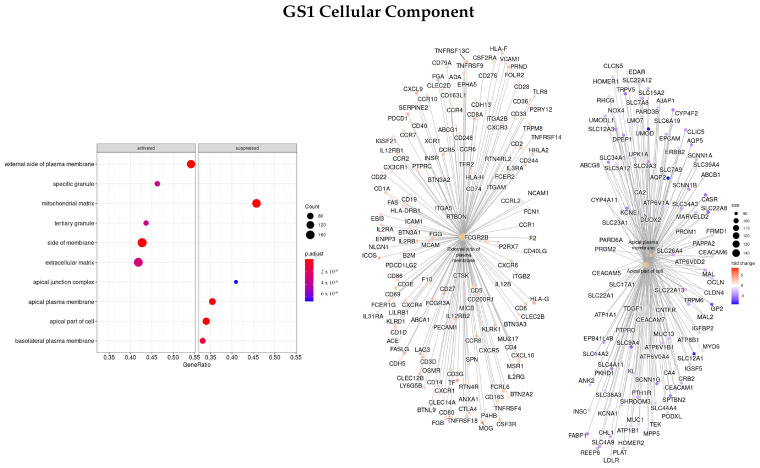
Gene ontology enrichment analysis regarding cellular components terms for a list of DEGs ranked by the log fold change between tumor and normal tissues. The left panel shows a dot chart with the most significant CC terms. The right panel shows a gene-concept network plot of the three most enriched terms that depicts the linkages of genes and biological concepts as a network.

**Figure 4 cells-11-02311-f004:**
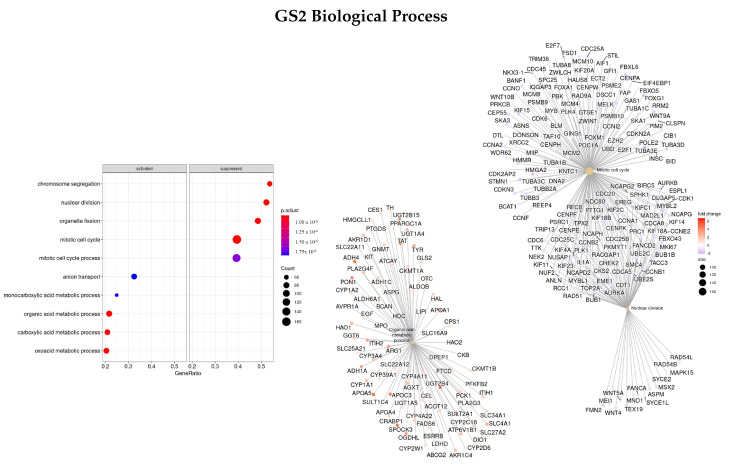
Gene ontology enrichment analysis regarding biological processes terms for a list of DEGs ranked by the log fold change between early and advanced stages of the disease. The left panel shows a dot chart with the most significant BP terms. The right panel shows a gene-concept network plot of the three most enriched terms that depicts the linkages of genes and biological concepts as a network.

**Figure 5 cells-11-02311-f005:**
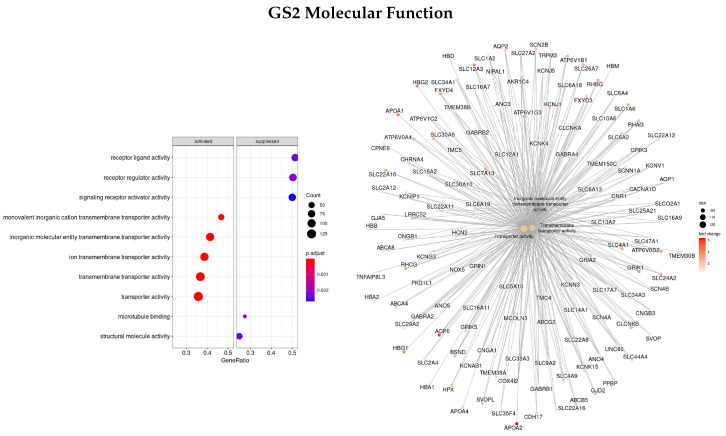
Gene ontology enrichment analysis regarding molecular functions terms for a list of DEGs ranked by the log fold change between early and advanced stages of the disease. The left panel shows a dot chart with the most significant MF terms. The right panel shows a gene-concept network plot of the three most enriched terms that depicts the linkages of genes and biological concepts as a network.

**Figure 6 cells-11-02311-f006:**
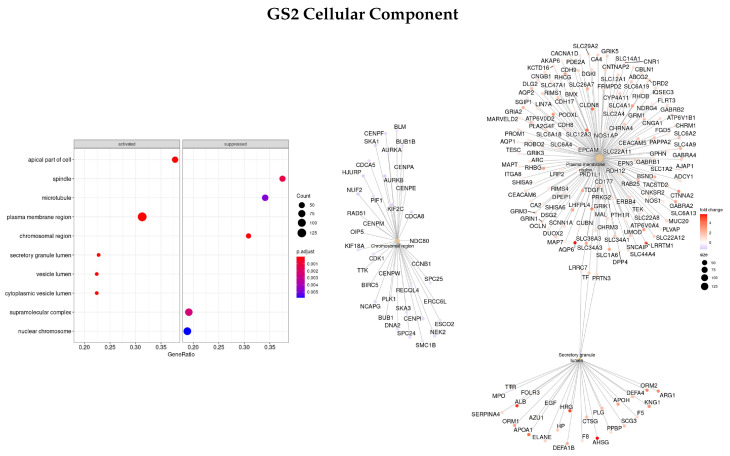
Gene ontology enrichment analysis regarding cellular components terms for a list of DEGs ranked by the log fold change between early and advanced stages of the disease. The left panel shows a dot chart with the most significant CC terms. The right panel shows a gene-concept network plot of the three most enriched terms that depicts the linkages of genes and biological concepts as a network.

**Figure 7 cells-11-02311-f007:**
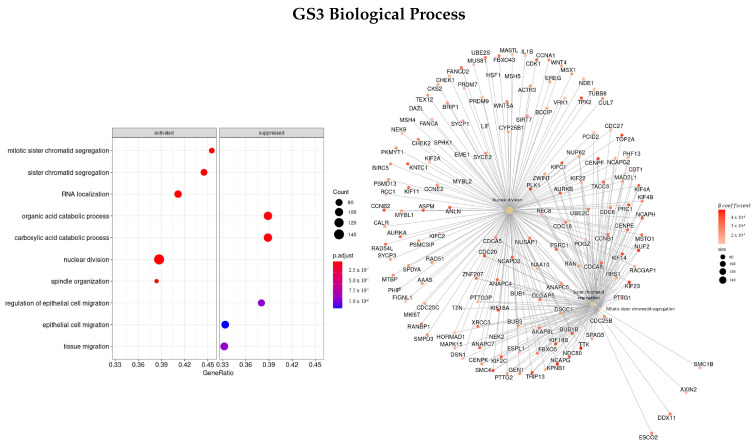
Gene ontology enrichment analysis regarding biological processes terms for a list of genes selected by EN ranked by the β coefficients of the regression. The left panel shows a dot chart with the most significant BP terms and on the right a gene-concept network plot of the three most enriched terms depicts the linkages of genes and biological concepts as a network.

**Figure 8 cells-11-02311-f008:**
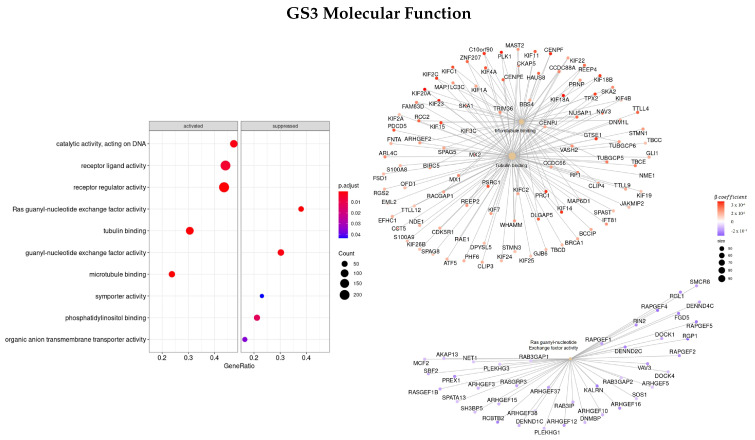
Gene ontology enrichment analysis regarding molecular function terms for a list of genes selected by EN ranked by the β coefficients of the regression. The left panel shows a dot chart with the most significant MF terms and on the right a gene-concept network plot of the three most enriched terms depicts the linkages of genes and biological concepts as a network.

**Figure 9 cells-11-02311-f009:**
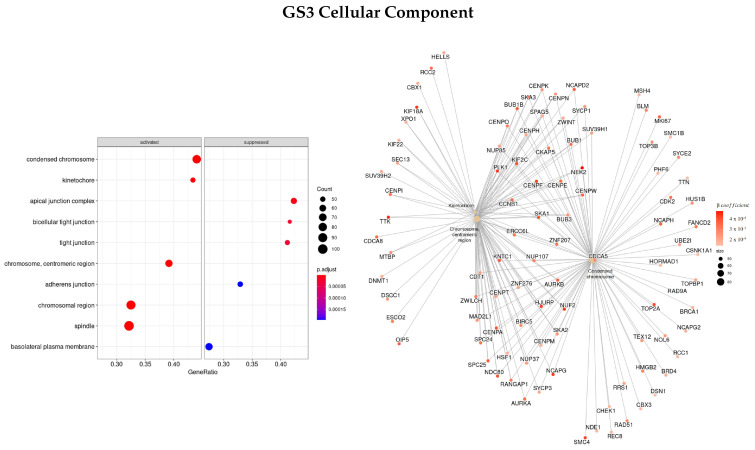
Gene ontology enrichment analysis regarding cellular components terms for a list of genes selected by EN ranked by the β coefficients of the regression. The left panel shows a dot chart with the most significant CC terms and on the right a gene-concept network plot of the three most enriched terms depicts the linkages of genes and biological concepts as a network.

**Figure 10 cells-11-02311-f010:**
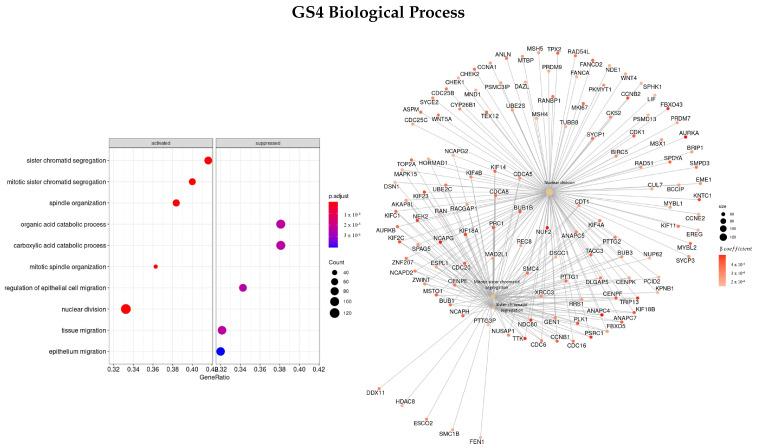
Gene ontology enrichment analysis regarding biological processes terms for a list of genes selected by TCox ranked by the β coefficients of the regression. The left panel shows a dot chart with the most significant BP terms. Right panel shows a gene-concept network plot of the three most enriched terms that depicts the linkages of genes and biological concepts as a network.

**Figure 11 cells-11-02311-f011:**
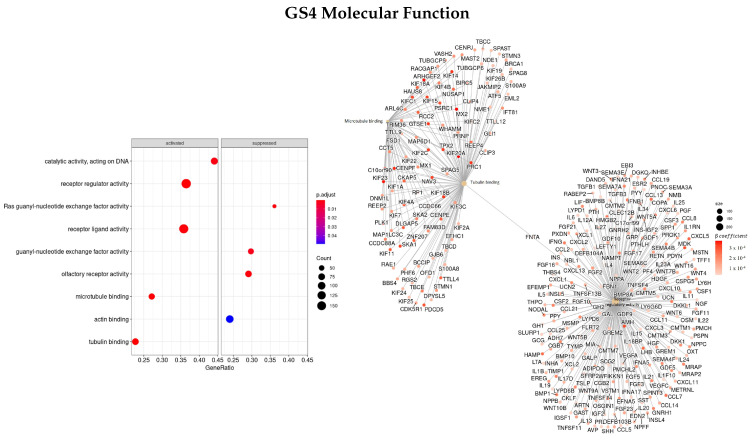
Gene ontology enrichment analysis regarding molecular function terms for a list of genes selected by TCox ranked by the β coefficients of the regression. The left panel shows a dot chart with the most significant MF terms. The right panel shows a gene-concept network plot of the three most enriched terms that depicts the linkages of genes and biological concepts as a network.

**Figure 12 cells-11-02311-f012:**
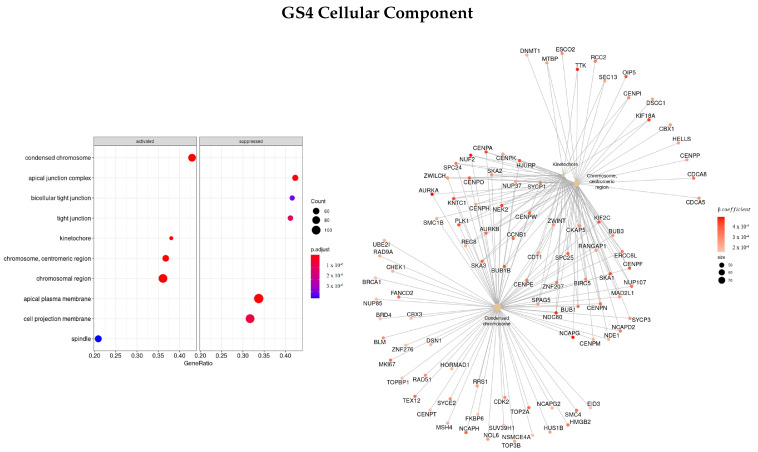
Gene ontology enrichment analysis regarding cellular components terms for a list of genes selected by TCox ranked by the β coefficients of the regression. The left panel shows a dot chart with the most significant CC terms. The right panel shows a gene-concept network plot of the three most enriched terms that depicts the linkages of genes and biological concepts as a network.

**Figure 13 cells-11-02311-f013:**
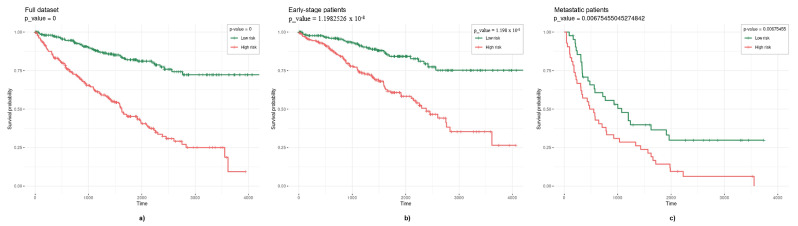
Kaplan–Meier curves obtained when applying a multivariate Cox model to a gene set comprising genes correlated with survival outcome in ccRCC and genes with some enriched ontology associated (p=24). (**a**) Full dataset (n=527); (**b**) early stage patients (n=441); (**c**) metastatic patients (n=84).

**Table 1 cells-11-02311-t001:** Data distribution regarding each clinical variable of interest: age (mean ± standard deviation), status (dead = 1 and alive = 0), stage (I, II, III and IV), T-stage (I, II, III and IV), N-stage (0 = not yet spread to nearby lymph nodes, 1 = spread to nearby lymph nodes), M-stage (metastasis = 1 and no metastasis = 0), sex (female and male) and race (Caucasian, African American, Asian).

	KIRC (n=537)	
**Age**	**60.57 **±** 12.15**	
Status	0	360 (67%)
	1	177 (33%)
Stage	I	269 (50%)
	II	57 (11%)
	III	125 (23%)
	IV	84 (16%)
T-stage	I	275 (51%)
	II	69 (13%)
	III	182 (34%)
	IV	11 (2%)
N-stage	0	240 (45%)
	1	17 (3%)
	x	280 (52%)
M-stage	0	426 (79%)
	1	79 (15%)
	x	30 (6%)
Sex	Female	191 (36%)
	Male	346 (64%)
Race	Caucasian	466 (87%)
	African American	56 (10%)
	Asian	8 (2%)
	*NA*	7 (1%)

**Table 2 cells-11-02311-t002:** Summary of the results obtained from Cox models using the two regularizations, EN and TCox. α controls the sparsity of the model (0.05<α<0.3). # genes—number of genes selected. All results are presented as mean values of the 100 runs tested.

α	0.3		0.2		0.1		0.05	
	* **p** * **-Value**	**# Genes**	* **p** * **-Value**	**# Genes**	* **p** * **-Value**	**# Genes**	* **p** * **-Value**	**# Genes**
EN	1.11 × 10^−18^	30	1.22 × 10^−17^	48	0	90	1.43 × 10^−16^	162
TCox	1.02 × 10^−15^	18	1.77 × 10^−15^	28	8.98 × 10^−16^	51	1.86 × 10^−15^	87

**Table 3 cells-11-02311-t003:** List of the ten most significant DEGs found between tumor and normal tissues from KIRC patients. LogFC—log fold change; FDR—false discovery rate.

Genes	LogFC	FDR
*MFSD4*	−5.34	1.96 × 10^−235^
*UNCX*	−6.42	2.46 × 10^−225^
*SEMG2*	−7.10	1.40 × 10^−195^
*RBP2*	−6.08	1.79 × 10^−183^
*GADL1*	−6.83	9.19 × 10^−179^
*LOC340094*	−5.68	1.23 × 10^−173^
*ELF5*	−8.18	6.98 × 10^−173^
*SIM2*	−4.52	9.04 × 10^−171^
*LOC284578*	−6.88	1.07 × 10^−170^
*HCRTR2*	−5.41	4.48 × 10^−169^

**Table 4 cells-11-02311-t004:** List of the ten most significant DEGs found between early (stages I, II and III) and advanced stages (stage IV) from KIRC patients. LogFC—log fold change; FDR—false discovery rate.

Genes	LogFC	FDR
*SPANXB2*	−5.70	2.27 × 10^−84^
*GABRA3*	−4.68	4.58 × 10^−67^
*MAGEC2*	−5.88	8.13 × 10^−57^
*SPATS1*	−4.41	7.53 × 10^−50^
*EPYC*	−4.09	6.73 × 10^−49^
*RDH8*	−4.20	6.43 × 10^−46^
*CSMD3*	−3.78	9.30 × 10^−42^
*BAAT*	−4.42	4.79 × 10^−39^
*TMEM158*	−2.46	3.31 × 10^−38^
*ANKFN1*	−3.35	2.34 × 10^−33^

**Table 5 cells-11-02311-t005:** List of the top 20 genes selected by elastic net in at least 50% of the runs when α=0.2. Arrows represent if genes are upregulated (↑) or downregulated (↓) in tumor tissue or in the advanced stage of the disease and % is the percentage of the runs where a certain gene appears in the solution. – genes that are not differentially expressed in tumor tissue.

Genes	%	DEGs Tumor Tissue	DEGs in Advanced Stage
*COPS7B*	100	↑	↑
*DONSON*	100	↑	↑
*LOC100272146*	100	↑	–
*CCNF*	99	↑	↑
*CKAP4*	99	↑	↑
*NCKAP5L*	99	↑	↑
*SEC61A2*	99	–	↑
*SNRPA1*	99	↑	↑
*STAT2*	99	↑	↑
*STRADA*	99	↓	↑
*NUMBL*	98	↑	↑
*SORBS2*	98	↑	↓
*CHFR*	97	↑	↑
*GIPC2*	96	↑	↓
*MBOAT7*	96	–	↑
*AR*	95	–	↓
*GTPBP2*	95	↑	↑
*KIF20A*	95	↑	↑
*NARF*	95	↑	↑
*FAM72B*	94	↑	↑

**Table 6 cells-11-02311-t006:** List of the 20 most selected genes by TCox when α=0.1. Arrows represent if genes are upregulated (↑) or downregulated (↓) in tumor tissue or the advanced stage of the disease and % is the percentage of the runs where a certain gene appears in the solution. – genes that are not differentially expressed in tumor tissue.

Genes	%	DEGs Tumor Tissue	DEGs in Advanced Stage
*GTF2E2*	100	↑	↑
*HAUS8*	100	↑	↑
*PRH2*	100	↑	↑
*SEC61A2*	100	–	↑
*ZNF18*	100	↑	↓
*C20orf72*	99	↑	↑
*DONSON*	99	↑	↑
*LOC286467*	99	↑	↑
*PRH1*	99	↑	–
*TFAP2E*	99	↑	↑
*TMEM86B*	99	↑	–
*TRAIP*	99	↑	↑
*DNAJC2*	96	↑	↑
*SLC26A6*	96	↓	↑
*SNRPA1*	96	↑	↑
*SORBS2*	96	↑	↓
*TCTE3*	96	↑	↑
*TMEM150C*	96	↓	↓
*C12orf32*	94	↑	↑
*C8orf44*	94	–	–

**Table 7 cells-11-02311-t007:** Genes most involved in the top three terms of each ontology (BP, MF and CC) for each gene set studied. GS1—logFC tumor vs. normal; GS2—logFC early vs. advanced stage; GS3—EN β coefficients; GS4—TCox
β coefficients;

GO	GS1	GS2	GS3	GS4
**BP**	*PAEP, ORM1, SAA1, LBP,* * AQP2, CALB1, NPHS1/2*	*TAT, CRABP1, APOA1/5, APOC3,* * UGT2B4, SULT2A1, ADH4, ANKFN1*	*SMC4, KIF, NEK2, PLK1,* * TTK, NUF2*	*NUF2, TRIP13, TTK, PRC1, KIF,* * NDC80, NEK2, PSRC1, AURKA, FBX*
**MF**	*AQP2, SLC12A1, ATP12A*	*APOA1/2, SLC12A3, AQP6*	*KIF, PLK1, PRC1, MX2,* * GTSE1, RGP1, TPX2*	*NAV3, HAUS8, IFNB1, CCL8,* * GTSE1, PSRC1, FNTA*
**CC**	*MUC17, SLC14A1/2, ATP12A,* * UMOD, AQP2*	*AHSG, TTR, HRG, ALB,* * AQP6, LRRTM2*	*NEK2, NUF2, TTK, PLK1,* * CENPF KIF*	*NCAPG, NEK2, TTK, OIP5,* * NUF2, PLK1, NDC80*

**Table 8 cells-11-02311-t008:** List of genes and corresponding β coefficients obtained in a multivariate Cox survival model using ridge regression. HR (hazard ratio) gives the effect size of covariates and it is calculated by exp(β). HR=1, no effect; HR<1, reduction in the hazard; HR>1, increase in the hazard.

Genes	Full Dataset		Early Stage		Advanced Stage	
	$\hat{\beta }$	HR	$\hat{\beta }$	HR	$\hat{\beta }$	HR
*COPS7B*	0.0930	1.10	0.0686	1.07	0.0539	1.06
*DONSON*	0.0840	1.09	0.0753	1.08	0.0557	1.06
*SEC61A2*	0.0731	1.09	0.0728	1.08	0.0552	1.06
*SNRPA1*	0.0481	1.08	0.0316	1.03	0.0226	1.02
*SORBS2*	−0.0977	0.91	−0.0688	0.93	−0.0616	0.94
*GTF2E2*	0.0687	1.07	0.0428	1.04	0.0271	1.03
*HAUS8*	0.0313	1.03	0.0240	1.02	0.0228	1.02
*PRH2*	0.0647	1.07	0.0734	1.08	0.0363	1.04
*ZNF18*	0.0427	1.04	0.0552	1.06	0.0251	1.03
*FBXO5*	0.0227	1.02	−0.0151	0.99	0.0258	1.03
*SMC4*	0.0544	1.06	0.0097	1.01	0.0491	1.05
*PRC1*	0.0053	1.01	0.0101	1.01	0.0210	1.02
*SLC34A1*	−0.0335	0.97	−0.0213	0.98	−0.0335	0.97
*SLC12A3*	−0.0166	0.98	−0.0111	0.99	−0.0115	0.99
*KIFC1*	−0.0045	1.00	−0.0082	0.99	0.0276	1.03
*KIF18A*	0.0187	1.02	0.0080	1.01	0.0445	1.05
*KIF23*	0.0121	1.01	0.0101	1.01	0.0315	1.03
*TPX2*	−0.0029	1.00	−0.0045	1.00	0.0415	1.04
*GTSE1*	0.0208	1.02	0.0063	1.01	0.0278	1.03
*SLC14A2*	0.0135	1.01	−0.0173	0.98	0.0381	1.04
*AQP2*	−0.0058	0.99	−0.0015	1.00	0.0090	1.01
*TTK*	0.0213	1.02	0.0046	1.00	0.0302	1.03
*NEK2*	0.0356	1.04	0.0392	1.04	0.0334	1.03
*NUF2*	0.0345	1.04	0.0263	1.03	0.0346	1.04

**Table 9 cells-11-02311-t009:** List of genes previously selected by both EN and TCox regularizers and corresponding β coefficients obtained when we applied a multivariate Cox survival model with ridge penalization. HR (hazard ratio) gives the effect size of covariates and it is calculated by exp(β). HR=1, no effect; HR<1, reduction in the hazard; HR>1, increase in the hazard.

Genes	Full Dataset		Early Stage		Advanced Stage	
	$\hat{\beta }$	HR	$\hat{\beta }$	HR	$\hat{\beta }$	HR
*DONSON*	0.1946	1.21	0.1920	1.21	0.2365	1.27
*SEC61A2*	0.1728	1.19	0.1857	1.20	0.2226	1.25
*SNRPA1*	0.1371	1.15	0.0857	1.09	0.0421	1.04
*SORBS2*	−0.2417	0.79	−0.1781	0.84	−0.2341	0.79

## Data Availability

Publicly available datasets were analyzed in this study. These data can be found here: https://portal.gdc.cancer.gov/projects/TCGA-KIRC (accessed on 1 May 2022). Furthermore, the code used to perform this analysis is available at https://github.com/sysbiomed/KIRC_analysis.

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
