# Peer review of "Kidney Cancer Biomarker Selection Using Regularized Survival Models"

_cells, 2022, doi:10.3390/cells11152311_

Round 1
Reviewer 1 Report
This is an interesting study regarding the biomarkers for cRCC using RNA-seq data.
There are some comments:
1: Regarding the Table 1, what the treatment modalities for these RCC patients? Surgery, Target therapy ( like VEGF inhibitor: sutent and votrient) or Immunotherapy or combined Target therapy and immunotherrapy?
since the effect of RNA expression profile on RCC survival may be different in RCC post surgery or in metastasis RCC post sutent/votrient or other therapy
2: if majority of the table 1 RCC case receive surgery, could author find the RNA expression data of " metastatic" RCC who receive target therapy or immunotherapy, and re-analysis the impact on survival?
3: functional study may be needed, or discuss
many biomarkers showed positive data, which one is more important, what the mechanism behind, through cell cycle regulators, metastasis, apoptosis?
4: these prognostic biomarkers for RCC, is only for early /clinical localized RCC , or also good biomarkers for metastatic RCC? why?
Reviewer 2 Report
The paper by Peixoto et al is a study that identify potential prognostic biomarkers correlated to survival outcome of ccRCC patients using two network-based regularizers (EN and TCox) applied to Cox model.
The study is well performed and very well written. It is an important and interesting application of the how to better find and evaluate potential prognostic biomarkers correlated to survival outcome of ccRCC patients. The result section is described in a way which is very easy to follow. The discussion part mention the biological function of the genes of interest.
Indeed, the use of molecular profiling in cancer diagnostics has expanded. This is an important contribution to the field of kidney cancer research and ccRCC.
However, there are are some minor points that need to be considered:
-A language check is needed
-Key words are general
-References 1, 7 and 25 are old; from 2008 and 2010, respectively. Are there more updated references?
-Is there a reference to the sentence: “Nowadays, there are no effective biomarkers for early diagnosis of RCC in clinical 36 treatment, and the molecular mechanism of RCC metastasis remains unclear" that can be added?
Round 2
Reviewer 1 Report
The raised questions and the limitations of current study were answered adequately.